# Pre- and Post-COVID-19 Antibiotic Consumption and Stewardship Program in a Southern Italian Hospital

**DOI:** 10.3390/antibiotics13121128

**Published:** 2024-11-24

**Authors:** Maria Costantino, Valentina Giudice, Federica Campana, Alessandra Anna Iannelli, Pasqualina Scala, Walter Longanella, Francesco Marongiu, Emilia Anna Vozzella, Maria Giovanna Elberti, Maria Alfieri, Giovanni Boccia, Valeria Conti, Francesco De Caro, Amelia Filippelli

**Affiliations:** 1Department of Medicine, Surgery, and Dentistry, University of Salerno, 84081 Baronissi, Italy; vgiudice@unisa.it (V.G.); f.campana@studenti.unisa.it (F.C.); pscala@unisa.it (P.S.); gboccia@unisa.it (G.B.); vconti@unisa.it (V.C.); fdecaro@unisa.it (F.D.C.); afilippelli@unisa.it (A.F.); 2University Hospital “San Giovanni di Dio e Ruggi d’Aragona”, 84121 Salerno, Italy; alessandra.iannelli@sangiovannieruggi.it (A.A.I.); walter.longanella@sangiovannieruggi.it (W.L.); direzione.sanitaria@sangiovannieruggi.it (E.A.V.); maria.elberti@sangiovannieruggi.it (M.G.E.); maria.alfieri@sangiovannieruggi.it (M.A.); 3Non-Profit Association F.I.R.S.Thermae, 80078 Pozzuoli, Italy; fmarongiu@firsthermae.org

**Keywords:** antibiotic stewardship, antibiotic consumption, COVID-19, antimicrobial resistance

## Abstract

Background/Objectives: Antibiotic resistance is a growing global threat that significantly impacts public health and healthcare costs. The Italian National Action Plan on Antimicrobial Resistance (PNCAR) was introduced in 2017 to address this issue by improving antibiotic stewardship. This study aimed to evaluate the effectiveness of the PNCAR in enhancing antibiotic management practices in a hospital in southern Italy before and after its implementation. Methods: We conducted an observational monocentric study to analyze antibiotic consumption in a hospital setting before and after the COVID-19 pandemic (2019 and 2023) and to examine prescription appropriateness and the types of used antibiotics. Results: After PNCAR introduction, we recorded a significant increase in antibiotic prescription appropriateness and in Access antibiotic and targeted therapy usage, while Reserve antibiotics were prescribed in ~10% of the cases, with an increasing trend in 2023. Conclusions: Our study supports the importance of targeted stewardship initiatives, including continuous monitoring and education, to sustain antibiotic prescription appropriateness and to reduce antimicrobial resistance.

## 1. Introduction

Antibiotic resistance has been identified by the World Health Organization (WHO) as one of the greatest threats to global public health, based on worldwide evidence of an exponential increase in multi-resistant pathogens (MDROs) [1,2]. Healthcare Stewardship promotes the appropriate, effective, and responsible use of available resources to enhance the quality of care, optimize patient outcomes, and support the sustainability and quality of the healthcare system. Antibiotic Stewardship (AS) programs consist of various practices and strategies for the proper use of antibiotics to improve clinical outcomes, minimize side effects, reduce antibiotic resistance, and efficiently manage financial resources. Indeed, antibiotic misusage increases healthcare costs due to longer hospitalization times and the employment of expensive drugs to treat MDROs [1]. The financial burden of antibiotic resistance also translates into a social burden with a loss of workdays for the affected subjects or for their caregivers. Therefore, a “One Health” approach involving both healthcare professionals and patients is needed for spreading education to increase the correct and appropriate use of the available drugs and for training healthcare professionals to ensure the adoption of the best clinical practices to reduce risk factors promoting MDRO infections [3,4,5].

Italy is the ninth Country with the highest antibiotics consumption, and antimicrobial resistance in Italy has a higher prevalence than in other European Countries, with an estimated economic impact of about EUR 320 million [4,5]. For these reasons, the Italian Government has developed the National Action Plan on Antimicrobial Resistance (PNCAR 2022–2025) for reverting this trend [6,7,8]. In these guidelines, appropriate antibiotics must only be prescribed when necessary, selecting the most suitable medications and ensuring the correct dosages and optimal treatment durations. Moreover, educational campaigns are promoted for the education of healthcare professionals, patients, and the general population on proper antibiotic use and the risks associated with antimicrobial resistance [9]. The continuous monitoring of antibiotic usage, coupled with feedback from the prescribing doctors, is another crucial aspect for improving prescription appropriateness. Prevention strategies, such as vaccinations, are also reinforced to reduce the incidence of infectious diseases and, eventually, the need for antibiotic-based treatments.

In 2023, the European Union Council has strongly recommended including antibiotic consumption monitoring in hospital settings, as a tool to monitor prescription practices and MDRO infections. Based on these recommendations, in this retrospective monocentric study, we describe antibiotic consumption in a southern Italian hospital before and after the COVID-19 pandemic. Indeed, the PNCAR was first established in 2017 and then updated after the pandemic in 2022 (plan 2022–2025), and we document the management changes during this period to ensure cost-effective therapies.

## 2. Results

### 2.1. Antibiotic Consumption

During the COVID-19 pandemic, hospitals managing severe SARS-CoV-2 infections faced difficulties to concretely follow the National Government Guidelines for antibiotic usage [7,10,11]. To better understand the evolution of hospital antibiotic employment before and after the COVID-19 pandemic and the effects of PNCAR indications in clinical practice, overall antibiotic consumption, expressed as Defined Daily Doses (DDDs), and usage variations, expressed as Δ, were calculated in 2019 and 2023 (Table 1). In detail, a 10.2% increase in the total number of consumed DDDs was registered, rising from 273,360 DDDs in 2019 to 301,123 DDDs in 2023, with cephalosporins as the most prescribed antibiotics in both years, especially in 2023 (86,593 DDDs in 2019 vs. 93,946 DDDs in 2023) (Table 1 and Table 2). Among cephalosporins, ceftriaxone was the most used drug in 2019, with 42,695 DDDs consumed, while piperacillin/tazobactam became predominant in 2023, with 49,053 DDDs, with a marked growth (+59.8%) (Table 2). Moreover, the total consumption of penicillin significantly increased to 43.7% in 2023, as well as that of amoxicillin/clavulanate (+24.4%) and ampicillin/β-lactamase inhibitors (+53.4%) (Table 1). This incrementation could reflect a strategic response to the rising prevalence of penicillin-resistant strains.

Overall, cephalosporins did not register a significant increase, showing only an 8.5% growth in 2023, while fifth-generation agents were on the rise. The increased use of ceftaroline fosamil (+503%) and ceftobiprole medocaril (+647.5%) between 2019 and 2023 may reflect a strategic response to the greater prevalence of resistant strains, particularly relevant during the COVID-19 pandemic with a higher incidence of secondary bacterial infections in hospitalized patients. Ceftaroline and ceftobiprole are effective against resistant pathogens, such as methicillin-resistant *Staphylococcus aureus* (MRSA) and *Streptococcus pneumoniae*, and are preferred options for critically ill patients with an increased risk of secondary or resistant pathogen infections [12]. Moreover, fifth-generation cephalosporins were preferred instead of fourth-generation cefepime (−73.9%), which is less effective against resistant pathogens. In 2023, we also observed the introduction of novel antibiotics, such as cefiderocol and aztreonam, designed to be effective against Gram-negative bacteria (Table 1).

International guidelines recommend reducing the use of fluoroquinolones, because of their side effects and increased risk of antimicrobial resistance, as well as of tigecycline, classified as a Reserve drug in the AWaRe system for tetracycline-resistant pathogens. In our hospital, we registered a downward trend for both fluoroquinolones (−27.0% for ciprofloxacin and −17.3% for levofloxacin) and tigecycline (−37.9%), consistent with previous studies [11]. Similarly, clindamycin (−54.3%), colistin (−50.4%), and glycopeptide antibiotics (vancomycin −39.4% and teicoplanin −57.3%) were less used in 2023 compared to 2019. Conversely, doxycycline, an Access drug in the AWaRe classification, was introduced, with consumption rising from 0 DDDs in 2019 to 705 DDDs in 2023 (Table 2), preferred over tigecycline. In addition, daptomycin (+192.9%) and linezolid (+195.3%) registered a significant increased consumption, likely because of a higher prevalence of Gram-positive resistant bacteria, such as MRSA and vancomycin-resistant *Enterococcus* (VRE), and complex infections in renal failure subjects. Fosfomycin is used to treat complicated urinary tract and/or resistant infections, as well as trimethoprim–sulfamethoxazole (TMP-SMX). In our study, both antibiotics were largely introduced in 2023 (fosfomycin, +893%; TMP-SMX, +29.8%), with corresponding reduction in sulfadiazine (−75%).

Carbapenem use raised from 24,413 DDDs in 2019 to 28,709 DDDs in 2023; however, meropenem use increased by 17.1%, while that of imipenem + cilastatin decreased by 30%, as well as that of ertapenem (a broad-spectrum carbapenem with limited coverage against certain Gram-negative bacteria; −62.8%). Meropenem/vaborbactam and imipenem + cilastatin and relebactam are newly introduced carbapenems combined with β-lactamase inhibitors to enhance efficacy against resistant bacteria and are increasingly used for the treatment of complex and resistant infections caused by beta-lactamase-producing bacteria. The introduction of these agents was also registered in our Institution, with 871 DDDs of meropenem/vaborbactam and 50 DDDs of imipenem + cilastatin and relebactam administered in 2023.

Amikacin is an aminoglycoside that was structurally modified to resist inactivation by certain bacterial enzymes and has broad activity against Gram-negative bacteria, including resistant strains. In our study, we observed a drastic reduction in the use of netilmicin (−99.6%), with an increased use of amikacin (+122.0%). In contrast, the complete elimination of chloramphenicol was documented, from 984 DDDs in 2019 to 0 DDDs in 2023, due to its known side effects and the availability of safer and more effective alternatives.

### 2.2. Antibiotic Consumption by AWaRe Category

In 2017, the WHO Expert Committee on Selection and Use of Essential Medicines developed the AWaRe classification of antibiotics to support AS, where drugs are divided in three groups, i.e., Access, Watch, and Reserve, based on their probability to induce antimicrobial resistance. The AWaRe system is also useful to monitor antibiotic consumption by classes and to evaluate the effects of AS on local hospital policies. Based on the 2019–2023 WHO 13th General Programme of Work, a country-level target of at least 60% of total antibiotic consumption must be reached for Access agents, as these drugs have a narrow spectrum of activity and a lower potential for antimicrobial resistance and are of lower cost compared to the other antibiotics, with a better safety profile [13,14]. Therefore, to explore antibiotic consumption and therapeutic strategies in our clinical and surgical departments and to monitor AS following the WHO 13th General Programme of Work criteria, the used antibiotics were divided by AWaRe category, and drug consumption expressed as DDD was compared between 2019 and 2023 [14,15,16]. In our Institution, Access antibiotics accounted for 34.1% (93,117 DDDs) of the total DDDs, with a slight increase (+15%) to 35.5% (106,866 DDDs) in 2023, far from the 60% goal. In contrast, Watch antibiotics, with broad-spectrum activity and a higher potential of developing resistance, represented the 58.3% (159,468 DDDs) of the total DDDs in 2019 and the 54.8% (164,907 DDDs) in 2023, with an almost stable use over the years (+3.4% increase). Moreover, we documented a marked increase (+41%) in Reserve antibiotics, last-resort drugs used for multidrug-resistant infections, with a high potential for antimicrobial resistance, from 7.6% (20,775 DDDs) of the total DDDs in 2019 to 9.7% (29,350 DDDs) in 2023.

### 2.3. Antibiotic Consumption by Medical Specialty

Next, we aggregated the data by medical, surgical, and emergency department, to evaluate the therapeutic strategies of different units and to highlight wards where AS should be better applied. Overall, the medical departments registered the highest antibiotic consumption increase (+16%) in 2023 compared to the surgical (+11%) and emergency units (+2.6%) (Figure 1). In detail, in the medical departments, cephalosporins were the most used antibiotics in 2019 and 2023 (22,213 and 24,000 DDDs, respectively), as well as penicillin (21,060 and 35,726 DDDs, respectively), followed by fluoroquinolones (11,901 in 2019 and 7289 DDDs in 2023) and carbapenems (9310 in 2019 and 11,812 DDDs in 2023) (Figure 1A). Similarly, in the surgical departments, cephalosporins and penicillin were the most used antibiotics in 2019 (42,253 and 14,959 DDDs, respectively) and 2023 (45,806 and 19,261 DDDs, respectively), followed by fluoroquinolones and carbapenems (8345 and 6022 DDDs in 2019, and 7032 and 6055 DDDs in 2023, respectively) (Figure 1B). In the emergency departments, cephalosporins and penicillin were the most employed antibiotics in 2019 (22,127 and 20,295 DDDs, respectively) and 2023 (22,008 and 25,933 DDDs, respectively), followed by carbapenems and nitroimidazoles (9081 and 7783 DDDs in 2019, and 10,842 and 8250 DDDs in 2023, respectively) (Figure 1C).

We further analyzed two units for each department, to monitor AS based on patient complexity, treated diseases, and internal guidelines in each division. For the medical departments, we analyzed the Hematology and Transplant Center, because of the presence of highly immunosuppressed and frail patients (e.g., transplanted subjects) and the Infectious Disease Unit, as “the model” for the clinical management of infectious diseases (Table 3). For the Hematology Unit, the total antibiotic consumption increased from 9922 DDDs in 2019 to 13,769 DDDs in 2023 (38.7%), because of a higher use of Access, such as amoxicillin and clavulanate, and Watch antibiotics, especially ceftriaxone, showing a good adherence to AS protocols, promoting the use of drugs with a lesser impact on resistance. Reserve drug consumption, such as that of vancomycin and linezolid, was stable, suggesting an effective control in the use of these drugs. In the Infectious Disease Unit, a significant increase in overall antibiotic consumption was documented, rising from 15,973 DDDs in 2019 to 20,638 DDDs in 2023 (+46.3%), with Reserve, such as linezolid and colistin, and Watch antibiotics, particularly piperacillin/tazobactam, as the most used drugs.

For the surgical departments, we considered the Orthopedics and Traumatology Unit and the General Surgery Unit (Table 4). In the Orthopedics and Traumatology Unit, antibiotic consumption slightly decreased (−2%) from 24,971 DDDs in 2019 to 22,468 DDDs in 2023, with an increased use of Watch antibiotics, such as levofloxacin. Reserve drugs were sparingly employed, suggesting improved management of postoperative infections by using more targeted therapies based on Watch antibiotics. In the General Surgery Unit, the total antibiotic consumption remained almost stable (+3.7%), varying from 9131 DDDs in 2019 to 9467 DDDs in 2023, with piperacillin/tazobactam as the most used antibiotic.

Among the emergency departments, the Emergency Surgery Unit and the Intensive Coronary Care Unit (ICU) were analyzed (Table 5). In the Emergency Surgery Unit, antibiotic consumption raised from 2935 DDDs in 2019 to 3142 DDDs in 2023, with Watch antibiotics as the most employed, especially piperacillin/tazobactam and ceftriaxone, while Reserve antibiotics, like vancomycin, showed limited use, despite the urgency for empirical treatments. In the ICU, antibiotic consumption slightly increased from 2935 DDDs in 2019 to 3142 DDDs in 2023, with a great use of Watch antibiotics, such as piperacillin/tazobactam, while the use of Reserve drugs, like linezolid and colistin, remained controlled.

Overall, these findings highlighted a significant shift in antibiotic prescribing patterns across various departments, with Watch antibiotics as the most prescribed, and a cautious approach to Reserve drugs. These trends suggest improvements in the clinical management of infections and the adherence to AS principles, promoting the use of effective antibiotics while mitigating the risk of resistance development.

## 3. Discussion

Inappropriate antibiotic use has led to a worldwide increase in infections caused by antibiotic-resistant bacteria [17]. Antimicrobial Stewardship Programs play a crucial role in fighting the antimicrobial resistance raise, as they promote, among other goals, antibiotic use optimization. Indeed, antibiotics must only be prescribed when truly necessary, at the correct dose and for the appropriate duration, to improve clinical outcomes and to minimize antimicrobial resistance. These programs ultimately promote patient safety and high-quality clinical management. A series of interventions can be implemented, including the monitoring of antibiotic use and the training of healthcare professionals. In our retrospective study, we focused on antibiotic prescription monitoring at a southern Italian hospital from 2019 to 2023—over the COVID-19 pandemic period, which has significantly impacted antimicrobial use.

At our institution, an overall antibiotic consumption increase was documented, similar to that reported nationwide in Italy, as the southern regions showed a 9.2% incrementation vs. our 10.2% increase [18]. To contrast this antibiotic misusage, the Italian Medicines Agency has discouraged the use of fluoroquinolones, due to a high risk of disabling, long-lasting, and potentially irreversible side effects [19], while encouraging the utilization of Access antibiotics. In compliance with Italian and European guidelines, penicillin combined with β-lactamase inhibitors, such as piperacillin/tazobactam and amoxicillin/clavulanate, were the most frequently prescribed antibiotics at our hospital, while a 22.9% reduction in fluoroquinolone use was observed, as well as a decrease in tigecycline utilization. These changes suggest a positive trend toward more responsible antibiotic practices and a more pathogen-specific management approach. For example, tigecycline should be reserved for tetracycline-resistant infections, to preserve its efficacy and reduce the risk of antimicrobial resistance.

Infections caused by Gram-negative bacteria, including *Klebsiella pneumoniae*, *Acinetobacter* species, *Pseudomonas aeruginosa*, and *Enterobacter* species, are of difficult management, because of the worldwide rise in multi-drug-resistant infections [20]. In these cases, polymyxins, including colistin, are widely used, with a pooled prevalence of colistin resistance of 3.1%, and the highest prevalence (up to 12.9%) in East Asia [21]. At our hospital, we registered a reduced use of colistin (−50.4%), reflecting a positive indication of more effective antibiotic management and greater attention to resistance prevention. Similarly, decreased utilization of vancomycin (−39.4%) and teicoplanin (−57.3%) and the complete abolishment of chloramphenicol were also documented. Vancomycin is the first-line drug for the treatment of MRSA; however, vancomycin-resistant pathogen (e.g., *Enterococcus faecium*) infections have more than doubled in the last years (up to 28.2% prevalence in 2021) [22]. Therefore, AS programs must promote the appropriate use of these antibiotics, and the reductions observed at our institutions are encouraging, as AS could be effective in optimizing antibiotic prescription appropriateness. Netilmicin use also decreased (−99.6%), replaced by safer and more effective options, like amikacin (+122.0%), in accordance with AS program recommendations. Conversely, we registered an increase in fifth-generation cephalosporins and certain Reserve antibiotics, especially in the medical and surgical departments. This trend could reflect both a greater spread of multi-resistant bacteria and the adoption of more targeted pharmacological approaches. However, these findings are alerts to intensify the AS program by enhancing the monitoring strategies, updating prevention protocols and guidelines for a prophylactic use, implementing protocols to prevent hospital-acquired infections, so to reduce the spread of MDROs.

To reduce the risk of adverse reactions and the development of bacterial resistance, the WHO categorizes antibiotics into three groups: Access, Watch, and Reserve. Access antibiotics should be used as the first-line treatment for most infections, with a target of at least 60% of total antibiotic consumption. Watch agents have a higher risk of inducing resistance and are generally recommended as second-line treatments or preferred only in specific cases. Reserve antibiotics are only intended for use in the most severe cases when all other options have failed [13,14]. In our study, from 2019 to 2023, a slight 14.8% increase in Access antibiotic consumption was documented, far from the WHO recommended 60%, although encouraging in a local context of AS program. The absolute consumption of Watch antibiotics showed a slight increase (+3.4%) in 2023 compared to 2019; however, the total DDDs consumed were reduced, dropping from 58.3% in 2019 to 54.8% in 2023. This reduction is a favorable sign, as it suggests a potential improvement in the prudent use of this category of antibiotics. Conversely, Reserve antibiotics were increasingly used, which likely reflects a possible raise in severe or pathogen-resistant infections.

## 4. Materials and Methods

### 4.1. Clinical Data

In this observational study, we monitored antibiotic consumption at the “Ruggi” hospital site of the University Hospital “San Giovanni di Dio e Ruggi d’Aragona”, Salerno, South Italy, before and after the COVID-19 pandemic. Antibiotics were classified using the Anatomical Therapeutic Chemical (ATC) Classification System [23,24]. The consumption data were sourced from the hospital pharmacy’s information systems. The antibiotic prescriptions from 1 January to 31 December 2019 were compared with those collected during the same period in 2023. Category and active molecules of consumed antibiotics in 2019 and 2023 were collected, and data from 2019 were used as a reference point, because that was the last year before the pandemic, while 2023 was the first post-pandemic year. Antibiotics were also classified using the WHO Access, Watch, Reserve (AWaRe) system [16,25,26].

### 4.2. Statistical Analysis

Data were analyzed using the MariaDB database, an open source “Relational Database Management System” (RDBMS) database based on a relational model that employs tables and relationships to store and organize data. The query language for data retrieval and manipulation was SQL (Structured Query Language), for efficient data management and complex querying. Antibiotic consumption was calculated as the number of antibiotic packs distributed per month to each hospital unit, and the data are expressed as defined daily dose (DDD), a statistical drug consumption measure defined by the WHO Collaborating Centre for Drug Statistics Methodology and used in combination with the ATC Code drug classification system for grouping related drugs. A *p* value < 0.05 was considered statistically significant.

## 5. Conclusions

This study highlights significant progress in the local antibiotic management after PNCAR introduction in 2017 in Italy, with a tendency of increasing prescription appropriateness, by preferring Access antibiotics where possible or specifically prescribing targeted drugs with appropriate activity spectrum. Reserve antibiotics were actually employed in 1/10 cases; however, there has been an increasing tendency in the last years, that could underlie a growing prevalence of complex and multi-resistant infections. To increase awareness and education for prescription appropriateness, we promote the training of healthcare professionals using clinical audits, interactive workshops, and prescription reviews with an AS expert, such as an infectious disease specialist, to ensure the adoption of best clinical practices and reduce the risk factors for MDRO infections. Our study reported only two aspects of AS programs (namely, antibiotic prescription appropriateness and training of healthcare professionals) and is the first step toward a full implementation of AS strategies in our local hospital. Therefore, continued monitoring and appropriate antibiotic prescription are two essential healthcare management strategies to prevent the development of antimicrobial resistance and ensure the sustainability of the available therapeutic options. However, AS programs should be fully implemented with the application of all seven core elements: leadership commitment; accountability; drug expertise; action; tracking; reporting; and education [27].

## Figures and Tables

**Figure 1 antibiotics-13-01128-f001:**
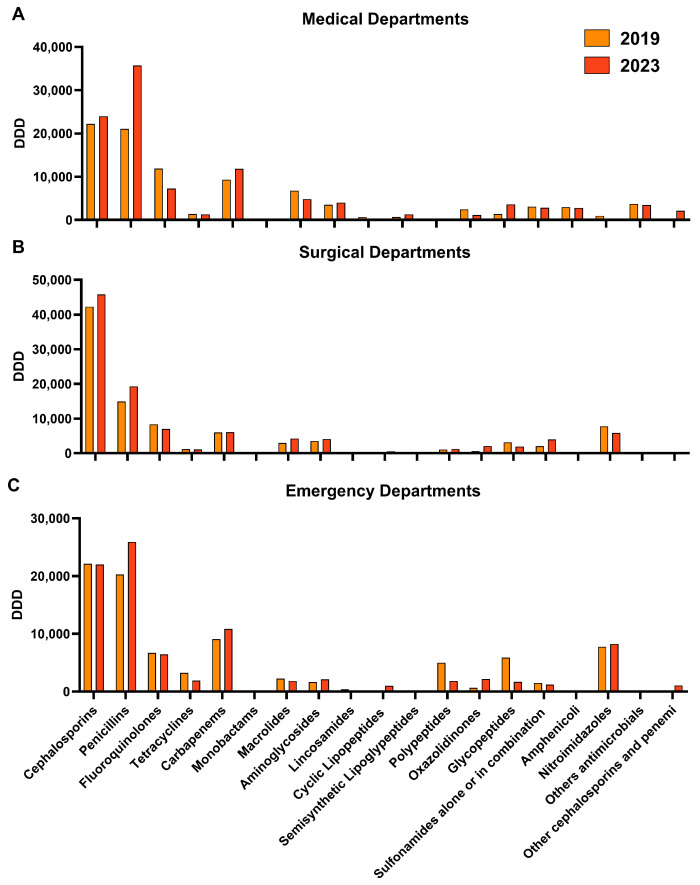
Antibiotic consumption expressed as Defined Daily Doses (DDD) in (**A**) medical, (**B**) surgical, and (**C**) emergency departments in 2019 and 2023.

**Table 1 antibiotics-13-01128-t001:** Antibiotic consumption expressed as Defined Daily Doses (DDD) and usage variations (Δ) between 2019 and 2023.

Class	2019 DDD	2023 DDD	*p* Value	Δ	% Usage 2019	% Usage 2023
Cephalosporins	86,593	93,946	<0.01	+8.5	31.7	31.2
Penicillin	56,314	80,920	<0.01	+43.7	20.6	26.9
Fluoroquinolones	26,948	20,785	<0.01	−22.9	9.9	6.9
Tetracyclines	5810	4312	<0.01	−25.8	2.1	1.4
Carbapenems	24,413	28,709	<0.01	+17.6	8.9	9.5
Monobactams	0	53	N.A.	N.A.	0	0.02
Macrolides	12,032	10,920	<0.01	−9.2%	4.4	3.6
Aminoglycosides	8667	10,281	<0.01	+18.6%	3.2	3.4
Lincosamides	1150	525	<0.01	−54.3%	0.4	0.2
Cyclic Lipopeptides	967	2832	<0.01	+192.9	0.35	1.0
Semisynthetic Lipoglypeptides	40	62	0.09	+55%	0.01	0.02
Polypeptides	8520	4227	<0.01	−50.4%	3.1	1.4
Oxazolidinones	2658	7849	<0.01	+195.3%	1.0	2.6
Other Antimicrobials	370	3674	<0.01	+893.0	0.1	1.2
Glycopeptides	12,126	6459	<0.01	−46.7%	4.4	2.2
Sulfonamides	6488	8002	<0.01	+23.3	2.4	2.7
Amphenicols	984	0	<0.01	−100%	0.36	0
Nitroimidazoles	19,280	17,567	<0.01	−8.9%	7.1	5.8
Total DDD	273,360	301,123		+10.2%		

Abbreviations: N.A., not applicable.

**Table 2 antibiotics-13-01128-t002:** Defined Daily Doses (DDD) of antibiotics, divided by class and specific active molecules, consumed in 2019 and 2023.

Class	Active Molecule	2019 DDD	2023 DDD	Δ
1st-Generation cephalosporins	Cefazolin	31,691	37,922	+19.7
2nd-Generation cephalosporins	Cefaclor	0	0	-
3rd-Generation cephalosporins	Ceftriaxone	42,695	42,329	−0.9
	Ceftazidime	4801	4713	−1.8
	Ceftazidime/avibactam	2010	2325	+15.7
	Cefotaxime	2711	2106	−22.3
	Ceftolozane + tazobactam	260	766	+194.6
	Cefpodoxime	154	133	−13.6
	Ceftibuten	18	0	−100
	Cefixime	0	66	NA
4th-Generation cephalosporins	Cefepime	2113	552	−73.9
5th-Generation cephalosporins	Ceftaroline fosamil	100	603	+503
	Ceftobiprole medocaril	40	299	+647.5
Other cephalosporins and penems	Cefiderocol	0	2132	NA
Broad-spectrum penicillin	Amoxicillin	5392	3530	−34.5
	Ampicillin	0	0	-
β-Lactamase-sensitive penicillin	Benzathine benzylpenicillin	222	173	−22.1
Penicillin combinations including β-lactamase inhibitors	Piperacillin/tazobactam	30,690	49,053	+59.8
	Ampicillin + β-lactamase inhibitors	11,270	17,289	+53.4
	Amoxicillin + clavulanate	8740	10,875	+24.4
Fluoroquinolones	Ciprofloxacin Levofloxacin	15,458 11,490	11,282 9503	−27.0 −17.3
Tetracyclines	Doxycycline	0	705	Na
	Tigecycline	5810	3607	−37.9
Carbapenems	Meropenem	23,170	27,124	+17.1
	Meropenem/vaborbactam	0	871	NA
	Imipenem + cilastatin	614	430	−30
	Ertapenem	629	234	−62.8
	Imipenem, cilastatin, relebactam	0	50	NA
Monobactams	Aztreonam	0	53	NA
Macrolides	Clarithromycin	10,338	6774	−34.5
	Azithromycin	1454	4036	+177.6
	Josamycin	240	98	−59.2
	Spiramycin	0	12	NA
Aminoglycosides	Gentamicin	6530	7237	+10.8
	Amikacin	1370	3041	+122.0
	Netilmicin	767	3	−99.6
Lincosamides	Clindamycin	1150	525	−54.3
Cyclic lipopeptides	Daptomycin	967	2832	+192.9
Semisynthetic lipoglypeptides	Dalbavancin	40	56	+40.0
	Oritavancin	0	6	NA
Polypeptides	Colistin	8520	4227	−50.4
Oxazolidinones	Linezolid	2658	7849	+195.3
Other antimicrobials	Fosfomycin	370	3674	+893.0
Glycopeptides	Vancomycin	7148 4978	4332 2127	−39.4 −57.3
	Teicoplanin			
Sulfonamides	Sulfadiazine	400	100	−75
	Sulfamethoxazole trimethoprim	6088	7902	+29.8
Amphenicols	Chloramphenicol	984	0	−100
Nitroimidazoles	Metronidazole	19,280	17,567	−8.9

Abbreviations: N.A., not applicable.

**Table 3 antibiotics-13-01128-t003:** Antibiotic consumption in two representative medical departments.

Department	Active Molecule	2019 DDD (%)	2023 DDD (%)
Hematology	Access	Access, 1951 (19.6%)	Access, 2509 (18.4%)
	Cefazoline	11 (0.11)	-
	Gentamicin	60 (0.60)	-
	Netilmicin	619 (6.2)	-
	Metronidazole	440 (4.4)	520 (3.78)
	TMP-SMX	821 (8.3)	1768 (12.8)
	Amoxicillin	-	20 (0.14)
	Amoxicillin + clavulanate	-	111 (0.81)
	Amikacin	-	90 (0.65)
	Watch	Watch, 6901 (69%)	Watch, 9363 (68.8%)
	Piperacillin + tazobactam	1790 (18.0)	3100 (22.5)
	Azithromycin	107 (1.1)	277 (2.01)
	Clarithromycin	216 (2.2)	306 (2.22)
	Levofloxacin	1620 (16.3)	2225 (16.2)
	Ciprofloxacin	248 (2.5)	439 (3.19)
	Meropenem	1750 (17.6)	1501 (10.9)
	Ceftazidime	37 (0.37)	128 (0.93)
	Ceftriaxone	530 (5.3)	404 (2.93)
	Teicoplanin	543 (0.55)	353 (2.56)
	Vancomycin	60 (0.60)	40 (0.29)
	Imipenem	-	590 (4.28)
	Reserve	Reserve, 1140 (11.4%)	Reserve, 1733 (12.7%)
	Colistin	370 (3.7)	260 (1.89)
	Linezolid	690 (9.9)	900 (6.54)
	Tigecycline	80 (0.81)	200 (1.45)
	Aztreonam	-	53 (0.38)
	Cefidericol	-	100 (0.73)
	Ceftazidime/avibactam	-	185 (1.34)
	Ceftolozano/tazobactam	-	35 (0.03)
Infectious Disease	Access	Access, 6940 (43.4%)	Access, 6068 (29.4%)
	Cefazoline	-	10 (0.05)
	Gentamicin	670 (4.7)	310 (1.49)
	Metronidazole	1017 (7.2)	290 (1.40)
	TMP-SMX	1507 (10.6)	1475 (7.11)
	Amoxicillin	60 (0.42)	12 (0.06)
	Amoxicillin + clavulanate	490 (3.46)	414 (2.0)
	Amikacin	180 (1.27)	40 (0.19)
	Ampicillin + sulbactam	2300 (16.2)	3310 (16.0)
	Benzylpenicillins	56 (0.40)	62 (0.30)
	Sulfadiazine	400 (2.8)	100 (0.48)
	Clindamycin	260 (1.83)	15 (0.07)
	Doxycycline	-	30 (0.14)
	Watch	Watch, 4764 (29.8%)	Watch, 9577 (46.4%)
	Piperacillin + tazobactam	820 (5.8)	4083 (19.7)
	Azithromycin	101 (0.71)	95 (0.46)
	Clarithromycin	148 (1.04)	182 (0.88)
	Levofloxacin	450 (3.18)	500 (2.41)
	Ciprofloxacin	70 (0.49)	35 (0.17)
	Meropenem	1110 (7.8)	2528 (12.2)
	Ceftazidime	280 (1.98)	416 (2.01)
	Ceftriaxone	618 (4.4)	1241 (5.98)
	Teicoplanin	159 (1.12)	30 (0.14)
	Vancomycin	853 (6.02)	414 (2.00)
	Ertapenem	155 (1.09)	7 (0.03)
	Cefepime	-	46 (0.22)
	Reserve	Reserve, 4269 (26.7%)	Reserve, 4993 (24.2%)
	Colistin	850 (6.0)	450 (2.17)
	Linezolid	2289 (16.2)	941 (4.54)
	Tigecycline	150 (1.06)	200 (0.96)
	Cefidericol	-	456 (2.20)
	Ceftazidime/avibactam	224 (1.58)	90 (0.43)
	Ceftolozano/tazobactam	80 (0.56)	-
	Cefataroline fosamil	40 (0.28)	280 (1.35)
	Ceftobiprole medocaril	20 (0.14)	220 (1.06)
	Dalbavancin	25 (0.18)	43 (0.21)
	Daptomycin	531 (3.75)	677 (3.26)
	Fosfomycin	60 (0.42)	1570 (7.57)
	Meropenem/vaborbactam	-	60 (0.29)
	Oritavancin	-	6 (0.03)

Abbreviations: TMP-SMX, trimethoprim–sulfamethoxazole.

**Table 4 antibiotics-13-01128-t004:** Antibiotic consumption in two representative surgical departments.

Department	Active Molecule	2019 DDD (%)	2023 DDD (%)
Orthopedics and Traumatology	Access	Access, 16,753 (69%)	Access, 16,344 (73%)
	Cefazoline	11,985 (49.4)	11,166 (49.7)
	Gentamicin	1220 (5.0)	1140 (5.1)
	Metronidazole	780 (3.2)	1060 (4.7)
	TMP-SMX	480 (2.0)	690 (3.1)
	Amoxicillin	555 (2.3)	154 (0.69)
	Amoxicillin + clavulanate	1257 (5.2)	854 (3.8)
	Amikacin	-	230 (1.02)
	Ampicillin + sulbactam	456 (1.9)	770 (3.4)
	Clindamycin	20 (0.08)	60 (0.27)
	Doxycycline	-	220 (0.98)
	Watch	Watch, 6623 (27%)	Watch, 5310 (24%)
	Piperacillin + tazobactam	830 (3.4)	597 (2.7)
	Azithromycin	58 (0.2)	44 (0.2)
	Clarithromycin	394 (1.6)	600 (2.7)
	Levofloxacin	500 (2.1)	325 (1.45)
	Ciprofloxacin	986 (4.1)	540 (2.4)
	Meropenem	890 (3.7)	800 (3.6)
	Ceftazidime	261 (1.1)	195 (0.87)
	Ceftriaxone	1290 (5.3)	1870 (8.3)
	Teicoplanin	84 (0.35)	189 (0.84)
	Vancomycin	1320 (5.4)	140 (0.62)
	Cefepime	10 (0.04)	10 (0.04)
	Reserve	Reserve, 895 (4%)	Reserve, 814 (4%)
	Colistin	370 (1.5)	80 (0.36)
	Linezolid	40 (0.16)	313 (1.39)
	Tigecycline	450 (1.9)	40 (0.18)
	Ceftazidime/avibactam	-	90 (0.40)
	Ceftolozano/tazobactam	-	60 (0.27)
	Dalbavancin	-	7 (0.03)
	Daptomycin	35 (0.14)	114 (0.51)
	Fosfomycin	-	110 (0.49)
General Surgery	Access	Access, 5471 (66%)	Access, 1900 (39%)
	Cefazoline	3016 (33.03)	904 (18.02)
	Gentamicin	140 (1.53)	220 (4.39)
	Metronidazole	1080 (11.8)	560 (11.16)
	TMP-SMX	80 (0.88)	66 (1.32)
	Amoxicillin	264 (2.89)	20 (0.40)
	Amoxicillin + clavulanate	160 (1.75)	105 (2.09)
	Amikacin	20 (0.22)	10 (0.20)
	Ampicillin + sulbactam	670 (7.34)	-
	Clindamycin	35 (0.38)	15 (0.30)
	Chloramphenicol	6 (0.07)	-
	Watch	Watch, 2500 (30%)	Watch, 2596 (53%)
	Piperacillin + tazobactam	890 (9.7)	1485 (29.6)
	Azithromycin	20 (0.22)	-
	Clarithromycin	78 (0.85)	90 (1.80)
	Levofloxacin	155 (1.7)	43 (0.86)
	Ciprofloxacin	228 (2.50)	206 (4.11)
	Meropenem	300 (3.3)	466 (9.29)
	Ceftazidime	330 (3.61)	49 (0.98)
	Ceftriaxone	1130 (12.4)	180 (3.59)
	Vancomycin	-	62 (1.24)
	Ertapenem	24 (0.26)	15 (0.30)
	Cefepime	155 (1.70)	-
	Reserve	Reserve, 350 (4%)	Reserve, 426 (9%)
	Colistin	140 (1.53)	59 (1.18)
	Linezolid	30 (0.33)	136 (2.71)
	Tigecycline	180 (1.97)	110 (2.19)
	Ceftazidime/avibactam	-	70 (1.40)
	Ceftolozano/tazobactam	-	40 (0.80)
	Fosfomycin	-	10 (0.20)
	Cefpodoxime	-	1 (0.02)

Abbreviations: TMP-SMX, trimethoprim–sulfamethoxazole.

**Table 5 antibiotics-13-01128-t005:** Antibiotic consumption in two representative emergency departments.

Department	Active Molecule	2019 DDD (%)	2023 DDD (%)
Emergency Surgery	Access	Access, 7669 (33%)	Access, 9179 (41%)
	Cefazoline	130 (0.6)	1780 (7.96)
	Gentamicin	-	215 (0.96)
	Metronidazole	5820 (24.8)	6075 (27.2)
	TMP-SMX	113 (0.5)	139 (0.62)
	Amoxicillin	-	36 (0.16)
	Amoxicillin + clavulanate	1112 (4.7)	478 (2.1)
	Amikacin	10 (0.04)	20 (0.09)
	Ampicillin + sulbactam	309 (1.3)	381 (1.7)
	Clindamycin	175 (0.7)	45 (0.20)
	Doxycycline	-	10 (0.04)
	Watch	Watch, 13,798 (59%)	Watch, 11,761 (53%)
	Piperacillin + tazobactam	4430 (18.9)	5127 (22.9)
	Azithromycin	3 (0.01)	28 (0.13)
	Clarithromycin	198 (0.8)	234 (1.0)
	Levofloxacin	855 (3.6)	480 (2.1)
	Ciprofloxacin	1564 (6.7)	1845 (8.2)
	Meropenem	2170 (9.2)	1280 (5.7)
	Ceftazidime	688 (2.9)	130 (0.58)
	Ceftriaxone	3310 (14.1)	2195 (9.8))
	Teicoplanin	290 (1.2)	242 (1.1)
	Ertapenem	95 (0.40)	35 (0.16)
	Vancomycin	195 (0.8)	145 (0.65)
	Cefepime	-	20 (0.09)
	Reserve	Reserve, 2010 (8%)	Reserve, 1424 (6%)
	Colistin	620 (2.6)	340 (1.5)
	Linezolid	50 (0.2)	174 (0.8)
	Tigecycline	1060 (4.5)	640 (2.9)
	Ceftazidime/avibactam	270 (1.2)	10 (0.04)
	Daptomycin	10 (0.04)	78 (0.35)
	Fosfomycin	-	30 (0.13)
	Cefiderocol	-	132 (0.59)
	Ceftaroline fosamil	-	20 (0.09)
Intensive Coronary Care Unit	Access	Access, 1475 (49%)	Access, 963 (32%)
	Cefazoline	1195 (40.7)	602 (19.2)
	Gentamicin	40 (1.4)	34 (1.08)
	Metronidazole	20 (0.68)	90 (2.9)
	TMP-SMX	-	42 (1.34)
	Amoxicillin	50 (1.7)	24 (0.76)
	Amoxicillin + clavulanate	100 (3.4)	6 (2.1)
	Amikacin		10 (0.32)
	Ampicillin + sulbactam	10 (0.34)	145 (4.6)
	Clindamycin	20 (0.68)	-
	Ampicillin	40 (1.4)	-
	Doxycycline	-	10 (0.32)
	Watch	Watch, 1362 (45%)	Watch, 1685 (55%)
	Piperacillin + tazobactam	160 (5.5)	550 (17.5)
	Azithromycin	-	53 (1.7)
	Clarithromycin	64 (2.2)	85 (2.7)
	Levofloxacin	220 (7.5	80 (2.5)
	Ciprofloxacin	112 (3.8)	36 (1.15)
	Meropenem	350 (11.9)	330 (10.5)
	Ceftazidime	24 (0.82)	56 (1.8)
	Ceftriaxone	230 (7.8)	470 (14.96)
	Vancomycin	106 (3.6)	18 (0.57)
	Teicoplanin	96 (3.3)	10 (0.32)
	Reserve	Reserve, 178 (6%)	Reserve, 403 (13%)
	Colistin	120 (4.1)	5 (0.16)
	Linezolid	-	45 (1.4)
	Tigecycline	10 (0.34)	30 (0.95)
	Ceftazidime/avibactam	20 (0.68)	50 (1.59)
	Ceftolozano/tazobactam	-	30 (0.95)
	Fosfomycin	10 (0.34)	110 (3.5)
	Daptomycin	8 (0.27)	58 (1.8)
	Ceftaroline fosamil	10 (0.34)	70 (2.2)
	Cefiderocol	-	1 (0.03)
	Ceftobiprole	-	4 (0.13)

Abbreviations: TMP-SMX, trimethoprim–sulfamethoxazole.

## Data Availability

Data are available upon request by the authors.

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
