# Peer review of "Pre- and Post-COVID-19 Antibiotic Consumption and Stewardship Program in a Southern Italian Hospital"

_antibiotics, 2024, doi:10.3390/antibiotics13121128_

Round 1

Reviewer 1 Report

Comments and Suggestions for Authors

This is an interesting retrospective study on antibody consumption, its association with AMR and the effect of antimicrobial stewardship programmes.

However the data reported in this study would become more solid and meaningful if they were to be coupled with analogous data for MDRO isolated in the hospital.

The authors report an overall increase in antibiotic consumption from the ID Department. However, corresponding DDDs are significantly reduced in 2023.

What was the isolation frequency of the different pathogens? Eg Acinetobacter was significantly less covered post covid, as both colistin and tigecyclin consumption were reduced.

English language of the manuscript needs intensive editing throughout.

Comments on the Quality of English Language

Severe editing is needed throughout the manuscript, both in terms of grammar and syntax.

Author Response

Comments and Suggestions for Authors

This is an interesting retrospective study on antibody consumption, its association with AMR and the effect of antimicrobial stewardship programmes. However, the data reported in this study would become more solid and meaningful if they were to be coupled with analogous data for MDRO isolated in the hospital.

What was the isolation frequency of the different pathogens? Eg Acinetobacter was significantly less covered post covid, as both colistin and tigecyclin consumption were reduced.

Response to General Comments. We thank the Reviewer for these valuable comments and suggestions. We are pleased that you have found our study of interest and methodologically sound. We completely agree with the Reviewer that data on multidrug-resistant organisms (MDRO) would be of great importance to strengthen our findings. However, the Antimicrobial Stewardship Program has been only recently implemented at our hospital, and MRDO monitoring together with systematic collection of comparable microbiological data are still ongoing. Indeed, we have started to track infections caused by Klebsiella oxytoca producing extended-spectrum beta-lactamases (ESBL), Klebsiella pneumoniae, and Stenotrophomonas maltophilia, with data showing their presence as early as 2019. However, these observations are still in the initial phase. We understand that this is a limitation of our study, however we hope that our results can still contribute to the literature on antimicrobial stewardship and antibiotic consumption, especially in settings where these programs are in the initial stages of implementation.

We have also added the following text to our conclusions.

This study highlights significant progress in local antibiotic management after PNCAR introduction in 2017 in Italy, with a tendency of increasing prescription appropriateness, by preferring Access antibiotics where possible or specifically prescribing targeted drugs with appropriate activity spectrum. Reserve antibiotics are actually employed in 1/10 of cases, however there is an increasing tendency in last years, that could underlie growing prevalence of complex and multi-resistant infections. To in-crease awareness and education for prescription appropriateness, we promoted training of healthcare professionals using clinical audits, interactive workshops, and pre-scription review with an AS expert, such as an infectious disease specialist, to ensure the adoption of best clinical practices and reduce risk factors for MRDO infections. Our study reported only two aspects of AS programs (namely, antibiotic prescription appropriateness and training of healthcare professionals) and is the first step toward a full implementation of AS strategies in our local hospital. Therefore, continued monitoring and appropriate antibiotic prescription are two essential healthcare management strategies to prevent development of antimicrobial resistance and ensure sustainability of available therapeutic options. However, AS programs should be fully implemented with the application of all seven core elements: leadership commitment; accountability; drug expertise; action; tracking; reporting; and education [27].”

Comment 1. The authors report an overall increase in antibiotic consumption from the ID Department. However, corresponding DDDs are significantly reduced in 2023.

Response to Comment 1. We thank the Reviewer for this point that should be clarify.

On page 6, lines 195-198, the following text is present “In the Infectious Diseases Unit, a significant increase in overall antibiotic consumption was documented, rising from 15,973 DDD in 2019 to 20,638 DDD in 2023 (+46.3%), with Reserve, such as linezolid and colistin, and Watch antibiotics, particularly piperacillin/tazobactam, as the most used.”

In Table 3, Access antibiotics are 6940 (43.4%) in 2019 and 6068 (29.4%) in 2023 (this might be the reduction reported by the Reviewer). However, Watch agents are 4764 (29.8%) in 2019 and 9577 (46.4%) in 2023, and Reserve are 4269 (26.7%) in 2019 and 4993 (24.2%) in 2023 (total 2019, 15973; and total 2023, 20638).

Comment 2. English language of the manuscript needs intensive editing throughout.

Response to Comment 2. We thank the Reviewer for this comment and we have checked English language throughout the manuscript.

Reviewer 2 Report

Comments and Suggestions for Authors

A respectful greeting, I appreciate the invitation to evaluate the paper, the information is correct and has the appropriate methodology, the graphs and tables are clear and allow the information to be reviewed quickly and in my opinion they are very complete, The study shows of the local experience on the rational use of antibiotics program and some outcomes before and after the pandemic, however, given that the information remains merely in the descriptive part, the conclusions are subject to the local reality and it is difficult to extrapolate to other geographical areas, it seems to me that the discussion should be deeper and provide elements to replicate the experience in other countries. 

I suggest correlating the DDD information with the antimicrobial resistance indicators for some specific scenarios since the information from before and after the pandemic could clarify some trends in consumption

Author Response

Comments and Suggestions for Authors

A respectful greeting, I appreciate the invitation to evaluate the paper, the information is correct and has the appropriate methodology, the graphs and tables are clear and allow the information to be reviewed quickly and in my opinion they are very complete, The study shows of the local experience on the rational use of antibiotics program and some outcomes before and after the pandemic, however, given that the information remains merely in the descriptive part, the conclusions are subject to the local reality and it is difficult to extrapolate to other geographical areas, it seems to me that the discussion should be deeper and provide elements to replicate the experience in other countries. I suggest correlating the DDD information with the antimicrobial resistance indicators for some specific scenarios since the information from before and after the pandemic could clarify some trends in consumption.

Response to General Comments. We thank the Reviewer for these thoughtful evaluation and constructive suggestions regarding our work.

Comment 1. The study shows of the local experience on the rational use of antibiotics program and some outcomes before and after the pandemic, however, given that the information remains merely in the descriptive part, the conclusions are subject to the local reality and it is difficult to extrapolate to other geographical areas, it seems to me that the discussion should be deeper and provide elements to replicate the experience in other countries.

Response to Comment 1. We greatly appreciate these Reviewer’s comments, and we have expanded the discussion section, delving deeper into the results and highlighting key aspects that could be adapted in other Countries. Discussion has been updated as follows.

Inappropriate antibiotic use has led to a worldwide increase in infections caused by antibiotic-resistant bacteria [17]. Antimicrobial Stewardship Programs play a crucial role to fight antimicrobial resistance raise, as they promote, among other goals, antibiotic use optimization. Indeed, antibiotics must be prescribed only when truly necessary, at the correct dose and for the appropriate duration, to improve clinical out-comes and to minimize antimicrobial resistance. These programs ultimately promote patient safety and high-quality clinical management. A series of interventions can be implemented, including monitoring of antibiotic use and training of healthcare professionals. In our retrospective study, we focused on antibiotic prescription monitoring at a Southern Italian hospital from 2019 to 2023— over the COVID-19 pandemic a period, that has significantly impacted on antimicrobial use.

At our institution, an overall antibiotic consumption increase was documented, similar to that reported nationwide in Italy, as the Southern regions have shown a +9.2% incrementation vs our +10.2% increase [18]. To contrast this antibiotic misusage, the Italian Medicines Agency has discouraged the use of fluoroquinolones, due to a high risk of disabling, long-lasting, and potentially irreversible side effects [19], while encourages the utilization of Access antibiotics. In compliance with Italian and Euro-pean guidelines, penicillin combined with β-lactamase inhibitors, such as piperacillin/tazobactam and amoxicillin/clavulanate, were the most frequently prescribed anti-biotics at our hospital, while a 22.9% reduction in fluoroquinolone use was observed, as well as a decrease in tigecycline utilization. These changes suggested a positive trend toward more responsible antibiotic practices and a more pathogen-specific management approach. For example, tigecycline should be reserved for tetracycline-resistant infections, for preserving its efficacy and for reducing the risk of anti-microbial resistance.

Infections caused by Gram-negative bacteria, including Klebsiella pneumoniae, Acinetobacter species, Pseudomonas aeruginosa, and Enterobacter species, are of difficult management, because of the worldwide rise of multi-drug-resistant infections [20]. In these cases, polymyxins, including colistin, are widely used in these cases with a pooled prevalence of colistin resistance of 3.1%, with the highest prevalence (up to 12.9%) in East Asia [21]. At our hospital, we registered a reduced use of colistin (-50.4%), reflecting a positive indication of more effective antibiotic management and greater attention to resistance prevention. Similarly, decreased utilization of vancomycin (-39.4%) and teicoplanin (-57.3%), and the complete abolishment of chloramphenicol were also documented. Vancomycin is the first-line drug for treatment of MRSA; however, vancomycin-resistant pathogen (e.g., Enterococcus faecium) infections have more than doubled in last years (up to 28.2% prevalence in 2021) [22]. Therefore, AS programs must promote the appropriate use of these antibiotics, and reductions observed at our institutions are encouraging, as AS could be effective in optimizing anti-biotic prescription appropriateness. Netilmicin use also decreased (-99.6%), replaced by safer and more effective options, like amikacin (+122.0%), in accordance with AS program recommendations. Conversely, we registered an increase in fifth-generation cephalosporins and certain “Reserve” antibiotics, especially in medical and surgical departments. This trend could reflect both a greater spread of multi-resistant bacteria and the adoption of more targeted pharmacological approaches. However, these findings are alerts to intensify AS program by enhancing monitoring strategies, updating prevention protocols and guidelines for prophylactic use, implementing protocols to prevent hospital-acquired infections, and reducing the spread of MRDO.

To reduce the risk of adverse reactions and the development of bacterial resistance, the WHO categorizes antibiotics into three groups: Access, Watch, and Re-serve. Access antibiotics should be used as the first-line treatment for most infections, with a target of at least 60% of total antibiotic consumption. Watch agents have a higher risk of inducing resistance and are generally recommended as second-line treatments or preferred only in specific cases. Reserve antibiotics are intended for use only in the most severe cases when all other options have failed [13,14]. In our study, from 2019 to 2023, a slight +14.8% increase in Access antibiotic consumption has been documented, far from the WHO recommended 60%, although encouraging in a local context of AS program. Absolute consumption of Watch antibiotics showed a slight increase (+3.4%) in 2023 compared to 2019; however, total DDD consumed were reduced, dropping from 58.3% in 2019 to 54.8% in 2023. This reduction is a favorable sign, as it suggests a potential improvement in the prudent use of this category of anti-biotics. Conversely, Reserve antibiotics were increasingly used, likely reflecting a pos-sible raise of severe or resistant-pathogen infections.”

Comment 2. I suggest correlating the DDD information with the antimicrobial resistance indicators for some specific scenarios since the information from before and after the pandemic could clarify some trends in consumption.

Response to Comment 2. We fully agree that correlating the DDD data with antimicrobial resistance indicators would provide a significant enhancement to our study. However, the Antimicrobial Stewardship Program has been only recently implemented at our hospital, and MRDO monitoring together with systematic collection of comparable microbiological data are still ongoing. Indeed, we have started to track infections caused by Klebsiella oxytoca producing extended-spectrum beta-lactamases (ESBL), Klebsiella pneumoniae, and Stenotrophomonas maltophilia, with data showing their presence as early as 2019. However, these observations are still in the initial phase. We understand that this is a limitation of our study, however we hope that our results can still contribute to the literature on antimicrobial stewardship and antibiotic consumption, especially in settings where these programs are in the initial stages of implementation.

We have also added the following text to our conclusions.

This study highlights significant progress in local antibiotic management after PNCAR introduction in 2017 in Italy, with a tendency of increasing prescription appropriateness, by preferring Access antibiotics where possible or specifically prescribing targeted drugs with appropriate activity spectrum. Reserve antibiotics are actually employed in 1/10 of cases, however there is an increasing tendency in last years, that could underlie growing prevalence of complex and multi-resistant infections. To in-crease awareness and education for prescription appropriateness, we promoted training of healthcare professionals using clinical audits, interactive workshops, and pre-scription review with an AS expert, such as an infectious disease specialist, to ensure the adoption of best clinical practices and reduce risk factors for MRDO infections. Our study reported only two aspects of AS programs (namely, antibiotic prescription appropriateness and training of healthcare professionals) and is the first step toward a full implementation of AS strategies in our local hospital. Therefore, continued monitoring and appropriate antibiotic prescription are two essential healthcare management strategies to prevent development of antimicrobial resistance and ensure sustainability of available therapeutic options. However, AS programs should be fully implemented with the application of all seven core elements: leadership commitment; accountability; drug expertise; action; tracking; reporting; and education [27].”

Round 2

Reviewer 1 Report

Comments and Suggestions for Authors

The authors have addressed the points raised adequately.

Reviewer 2 Report

Comments and Suggestions for Authors

Respectful greetings, I agree with the authors' clarifying comments, and I consider that it could be published in this form.